# Balanced Reporting and Boomerang Effect: An Analysis of Croatian Online News Sites Vaccination Coverage and User Comments during the COVID-19 Pandemic

**DOI:** 10.3390/vaccines10122085

**Published:** 2022-12-06

**Authors:** Željko Pavić, Adrijana Šuljok, Juraj Jurlina

**Affiliations:** 1Faculty of Humanities and Social Sciences, Josip Juraj Strossmayer University of Osijek, 31000 Osijek, Croatia; 2Institute for Social Research in Zagreb, 10000 Zagreb, Croatia

**Keywords:** vaccination, COVID-19, media, Internet, boomerang effect, balanced reporting, vaccine hesitancy

## Abstract

The purpose of this paper was to explore online media coverage of COVID-19 vaccination and user reactions to the different types of coverage. The authors aimed to investigate possible boomerang effects that arise when COVID-19 media coverage is assertive and confident, and to determine the effects of balanced reporting. A two-stage random sample comprised a total of 300 articles published in three Croatian online news sites during a period from 1 February 2020, through 15 January 2022. The data were categorized using human coding content analysis, while reliability of coding was checked by using two coders and calculating reliability coefficients. The data were analyzed by means of negative binomial regression analysis. The results revealed that COVID-19 reporting was mainly consensual, i.e., it provided largely affirmative information about vaccines. However, user comments were highly polarized and mostly negative, with the majority of anti-vaccination tropes linked to the “corrupt elites”. Based on the user comments, the negative influence of balanced reporting on COVID-19 vaccines and the existence of boomerang effect in cases of the overtly persuasive affirmative reporting was also established. The boomerang effect did not depend on the context, i.e., on the type of reporting. This study extends previous research on balanced reporting and boomerang effects by analyzing online comments as a potentially good parallelism of the offline discursive strategies of the pro-vaccination and anti-vaccination communication. The results of the study can be used for the adjustment of strategic communication targeting the vaccine hesitant audience. Based on the study results, it is recommended that relativization and politicization of science should be prevented by not equating scientific consensus with absolute epistemological certainty and by addressing legitimate concerns of vaccine hesitant persons without putting explicit blame on them.

## 1. Introduction

It has been estimated that only less than 0.013% of the total published scientific articles enjoy media presence, which means, that even in the Internet era, journalists still act as strong gatekeepers who direct the flow of scientific knowledge to the public [1]. The media does not cover all topics equally-media select, filter and interpret newsworthy topics because they do not have enough time and space [2]. Scientific topics increasingly become the subject of media interest as they are a part of broader social and political problems [3]. For instance, it has been shown that traditional and influential media outlets can have a direct persuasive effect on the science attitudes [4], while it can be assumed that the changing media environment and new forms of online science journalism might exert positive influence by providing the interested public with relevant sources of rapid science information [5]. However, issues of trust in experts and expert systems are concerns of great importance for contemporary societies. Giddens asserted that we rely increasingly on experts to manage risk, but also argued that indefiniteness of expert knowledge sometimes makes it difficult for people to have trust in experts [6]. The risk society is paradoxically characterized by increasing dependence on experts along with declining trust in these same experts [7], since new knowledge brings up questions among experts that most often cannot be resolved simply by gathering additional information; instead, the dilemmas deepen, widen and multiply [8].

Even before the current pandemic, the ideas about possible influences of the media on vaccine hesitancy had been brought forward [9,10]. In their systematic analysis of the studies which employed content analysis of traditional media vaccination coverage, Catalan-Matamoros and Peñafiel-Saiz found that most studies determined that the media were more often focused on negative aspects of vaccines [11]. For Kata, the Internet is a “postmodern Pandora’s box”, given that it provides a lot of material that enables a person to question scientific authority and legitimacy [12]. When it comes to social media, it has been established that negative tweets about vaccination receive more retweets than positive tweets [13,14], and that negative Instagram posts about vaccination receive more likes [15]. Blankenship et al. suggest that the providers of negative information are more often very committed to their cause, i.e., that the issue of vaccination is more salient to them than to those who accept vaccines [14]. This is also confirmed in other research which showed that users belonging to anti-vaccination communities consume more social media sources and are more active [16,17]. In an experimental study, Betsch et al. found that viewing vaccine-critical websites increased the perception of risk of receiving vaccines, while it decreased the perception of risk of omitting vaccination and the intentions to vaccinate [18]. It can be assumed that the negative role of the misinformation on the Internet became even more pronounced during the COVID-19 pandemic given that the situation was new and unfamiliar [19].

When it comes to the types of arguments provided by vaccination skeptics, a codebook developed by Hughes et al. [20] identified main anti-vaccination narrative tropes and rhetorical strategies to be used in subsequent research on COVID-19 vaccination. They defined narrative tropes as themes which are part of the wider worldview narratives that are implied in the manifest tropes, while rhetorical strategies are more specific devices that can be utilized in various narrative tropes in order to shape appealing and persuasive messages. They identified the following four most common narrative tropes on the US anti-vaccination social media accounts: Corrupt Elite (powerful elites disempower ordinary citizens), Vaccine Injury (health damages), Sinister Motives (shadowy motives) and Heroes and Freedom Fighters (civil rights issues). Additionally, the following four most often used rhetorical strategies are identified: (1) the Brave Truthteller, (2) Do Your Own Research, (3) Mountains and Molehills and (4) Global Movement/Sleeping Giants.

When it comes to the user reactions, the studies of science communication are focused on the ways in which various types of communication can elicit both positive and negative attitudes and behavior. In this study, we also specifically focus on two issues: balanced reporting and boomerang effects that stem from overly confident science communication. Journalistic norm of balanced reporting is often cited as an important cause of negative views about scientific consensus [21]. By conducting a media content analysis and a survey study related to the MMR debate, Lewis and Spears confirmed that the journalistic norm of balanced reporting created an impression that there are two bodies of conflicting evidence that are equally important [22]. The media coverage which emphasizes scientific dissensus over COVID-19 vaccination issues can occur in two main forms. The first one is related to the negative comments about vaccination and vaccines made by the “whistleblowers” scientists. The other way is related to the misunderstandings over the very nature of scientific consensus, which is never finished and continually evolves. For instance, Capurro et al. found that some journalists in Canada covered COVID-19 crisis by emphasizing “duelling experts” and sending mixed messages about official public health measures [23]. Part of this problem might be derived from the process within the science communication, wherein knowledge becomes solidified and translated into a “fact” [24]. The perception of the level of dissensus among scientists may be further raised by employing journalistic norms of balanced coverage, even in the cases where the weight of evidence is heavily skewed towards one side [25]. Balanced coverage helps journalists to emphasize controversy and add drama to their reporting and to provide them with a guise of objectivity [26]. Studies showed that balanced reporting is quite widespread. For instance, Boykoff and Boykoff found that the majority of US press accounts on global warming gave roughly equal attention to the opposing views [27]. Boykoff’s content analysis of the main US television channels showed that about 70% of all news stories included balanced coverage of anthropogenic climate change [28]. By analyzing the U.S. media reporting, Merkley found that fully balanced reporting is not a universal journalistic norm, but that is still present in 20% of all accounts [29]. The effect of the balanced reporting was confirmed in several experimental studies. For example, Koehler established that balanced conflicting comments from an expert on either side of an issue influenced that ability of people to distinguish between topics with high and low expert consensus, even in cases where weight of evidence comment was provided [25]. Bolsen and Druckman found that an experimentally induced scientific consensus statement increased belief in human-induced climate change [30]. Similarly, van der Linden et al. found that experimentally manipulated public perception of the scientific consensus translated into a significant increase in climate change concern [31]. Aklin and Urpelainen confirmed that even the smallest amount of scientific dissent undermined public support for new environmental regulations [32].

On the other hand, overwhelmingly positive strategic communication can cause the so-called boomerang effect. Byrne and Hart suggest that a boomerang effect can occur due to either intended or unintended construct activation [33]. The first path occurs when the receiver receives the message in the intended way, but nevertheless does not accept it. This often happens due to psychological reactance, which arises from the perceived threat to personal freedom and the willingness to restore freedom of choice [34]. On the other hand, unintended construct activation happens when the message is either too complex or not too salient for the receiver. Several research studies in the field of health communication demonstrated the existence of the boomerang effect. Namely, in some cases health communication aimed at decreasing unhealthy behavior backfired by producing boomerang effects [35,36,37]. As for the COVID-19 communication, Shoenberger, Kim and Sun found that threat messaging related to brand communication increased reactance, and consequently led to lower intention to engage in socially responsible COVID-19 related behaviors [38]. It can be expected that the boomerang effect can be very pronounced among those with vaccine conspiracy beliefs, given that the process of motivated reasoning should be very strong. Vivid reporting and the reporting which presents concrete and extreme cases and personal cases, are also more likely to be remembered and activated [39,40,41], probably by virtue of the availability heuristic [42]. Scientists have noticed the idea and use this type of communication. For instance, Holton et al. found that during the peak of MMR controversy scientists more often used opinions/points of view/personal experiences than “pure facts” [43]. However, on a more negative note, the vividness can amplify negative attitudes in cases of conflict and low trust. For instance, Van Prooijen demonstrated that existential threats stimulate conspiratorial thinking only in cases where antagonistic groups are present and salient [44]. In the case of COVID-19 communication, we can hypothesize that the presence of the concrete scientists or other experts should serve as a vivid identification of the antagonistic group, thus increasing negative attitudes.

Based on the above considerations, the goal of this study is fourfold. The first goal is to investigate the presence of various vaccination and anti-vaccination tropes and rhetorical strategies during the pandemics. This goal is largely descriptive and does not entail specific hypotheses. Second, our aim was to check whether “balanced” reporting will elicit negative reactions among the readers. Third, we aimed to investigate possible boomerang effects that arise when COVID-19 coverage is too assertive and confident. Fourth, we wanted to explore possible interaction between balanced reporting and predictors of boomerang effect. i.e., whether predictors of boomerang effect will act more strongly in cases of balanced reporting. We set out to explore these issues by analyzing user comments posted within online news sites, since they represent a good way of estimating real-life discursive strategies of pro-vaccination and anti-vaccination communication appeals [45]. As opposed to user responses on social media, the topic of online news sites’ user comments and their dependence on the nuances of science communication is largely unexplored. Having all this in mind, we put forward the following research question and hypotheses:

Research question RQ1: Which types of anti-vaccination and pro-vaccination tropes and rhetorical strategies are present within online news sites’ user comments?

**General Hypothesis** **H1.***Implied scientific dissent about COVID-19 vaccination will elicit more negative user comments*.

**General Hypothesis** **H2.***There will be a boomerang effect when reporting on COVID-19 vaccines*.

**Specific Hypothesis** **H2.1.***Explicit call for vaccination will elicit more negative user comments*.

**Specific Hypothesis** **H2.2.***Blaming of vaccine hesitant people will elicit more negative user comments*.

**Specific Hypothesis** **H2.3.***Explicit naming of the expert will elicit more negative user comments*.

**General Hypothesis** **H3.***Predictors of boomerang effect will be stronger in cases of balanced reporting*.

**Specific Hypothesis** **H3.1.***Explicit call for vaccination will interact with balanced reporting in eliciting more negative user comments*.

**Specific Hypothesis** **H3.2.***Blaming of vaccine hesitant people will interact with balanced reporting in eliciting more negative user comments*.

**Specific Hypothesis** **H3.3.***Explicit naming of the expert will interact with balanced reporting in eliciting more negative user comments*.

## 2. Materials and Methods

The overall research design comprised the steps delineated in Figure 1. Namely, after the initial coding book development and sampling, preliminary reliabilities were calculated, which led to the development of the final version of the coding book. After that, data collection and data analysis were conducted. All the steps are described below in more detail.

As a means of operationalization of the above-mentioned research question and hypotheses, we developed a coding book to be used in the subsequent human coding content analysis. One part of the coding book covered vaccination tropes and rhetorical strategies as described in Hughes et al. [20]. Additionally, positive, neutral and negative user comments are counted. In the other part, the specific types of issue coverage are measured, namely balanced reporting and coverage that could elicit a boomerang effect. Balanced reporting was operationalized as the implied scientific dissent on the topic of vaccination and was recorded on a 1 to 5 scale, with higher number indicating more divergence on opinions, on the basis of whether reporting on the topic was exclusively affirmative, i.e., if it presented only one side of the argument; or dissenting opinions were brought forth, most often on the basis of opposing research or the opinions of experts whose opinions were contrasted. It has been suggested that science journalists should supplement balanced reporting with the weight of evidence acknowledgement which should point out that the bulk of evidence and support of the majority of experts lie on one side of the debate [46]. However, in the current study the weight of evidence argument was completely absent, since journalists largely avoided expressing their opinion in the cases where they presented conflicted evidence and dissenting views among the scientific community. Therefore, it was not necessary to differentiate among different versions of balanced reporting, i.e., to code the weight of evidence.

As for the examination of boomerang effects, explicit calls for vaccination were present in cases where experts directly addressed the general audience and encouraged them to get the vaccine. Blaming of hesitant persons was measured as an implicit or explicit statement that the pandemic cannot be stopped unless the vaccination rate is increased, or a contention that vaccination has no alternative in the current public health situation. The measurement of the presence of the named expert was quite straightforward.

The study employed a two-stage sample design. In the first stage, we used a judgment sample by selecting three online news sites with publicly available user comments-Index.hr, 24 sata.hr and Vecernji.hr-which are among the top five most visited online newspapers in Croatia judged by the average daily real users, either according to the Gemius rating of Croatian web domains [47] or according to the Alexa rating [48]. Both 24 sata.hr and Vecernji.hr represent online versions of the national daily journals, while Index.hr is an online-only media outlet. All three newspapers publish in Croatian. In the second stage, we chose 100 articles published by each portal by using random sampling, thus obtaining the total sample size of 300. The keywords used in finding the suitable articles were (English translation): “COVID-19”, “coronavirus”, “SARS-CoV2”; “scientists”, “scientific research” and various combinations of these keywords. Following the initial sample selection, the screening was conducted in order to include only those reports which entailed sufficient coverage of COVID-19 vaccination issues. The time period in which the analyzed articles were published was from 1 February 2020 to 15 January 2022, while the data were collected during February and March 2022. The study approval was obtained from the Ethics Committee of Faculty of Humanities and Social Sciences, Josip Juraj Strossmayer University of Osijek (approval number: 2158-83-02-19-2). The data are available as a Appendix A to this paper (Appendix A).

After two independent coders finished the analysis on a random sample of 30 cases, preliminary reliability coefficients were calculated. Given that the reliabilities below the lowest threshold value of 0.667 for the anti-vaccination tropes, major antagonists and rhetorical strategies were obtained, some of the categories were merged [49]. Namely, several categories of anti-vaccination tropes are merged into (a) corrupt elites and (b) vaccine injury. Similarly, major antagonists were merged into (a) mainstream society at large/politicians; shadowy villains and (b) the vaccine itself. Anti-vaccination rhetorical strategies were merged into (a) The Brave Truthteller; Global Movements/Sleeping Giants and (b) Do Your Own Research; Mountains and Molehills. Finally, pro-vaccination rhetorical strategies were merged into (a) Rationalism against irrationalism and (b) Presenting evidence for vaccination. After the mergers, final reliability calculations on a total sample of 80 cases were obtained. From Table 1 it can be seen that even after the merger of the categories, reliability of anti-vaccination tropes and major antagonists still remained slightly below the above-mentioned threshold.

## 3. Results

In Figure 2, the averages of comment types (affirmative, negative and neutral) cross-tabulated across the online news sites are shown. We can observe that the average number of comments per article across all the newspapers is about 66, and that the majority of comments are negative (40.17 per article).

The main tropes, antagonists and rhetorical strategies are shown in Table 2. We can behold that the majority of tropes are connected to the notion of corrupt elites and, consequently, that the major antagonist is not the vaccine itself, but politicians, mainstream society at large and the undefined shadowy villains. Within the tropes, the major anti-vaccination rhetorical strategy is linked to the “do your own research” approach which implies that the pandemics is an artificially and sometimes intentionally produced “mountain out of the molehill”. Regarding the pro-vaccination tropes, in the large majority of cases the proponents of COVID-19 vaccination simply pointed to the fact that vaccines work and prevent the disease. However, when doing that, in most cases the proponents implied irrationality of the vaccine hesitancy and refusal, thus indicating the high degree of polarization that could be noted in the comments.

Descriptive statistics for the variables related to the issue coverage characteristics are shown in Table 3. We can observe that in 71.24% of all cases the names of the experts are explicitly mentioned, while in the remaining cases only general findings and arguments are discussed, sometimes only with a mention of institutions where research touched upon had taken place. The take on vaccination in the analyzed articles is mainly consensual, given that only 7% of cases presented vaccination as mainly or completely controversial. Maybe surprisingly, the implicit or explicit blaming of vaccine hesitant people was present in 74.33% of all cases. However, in most cases the blame was quite implicit, most often in the form of the assertion that the pandemics cannot be stopped if people keep refusing to get vaccinated. Explicit call for vaccination was present in 54% of all cases.

In order to test the proposed hypotheses, we used several negative binomial regressions as an analytical strategy. The rationale for using negative binomial regression instead of ordinary least squares regression or Poisson regression was the fact we had count outcome variable and that there was a significant amount of over-dispersion (variance of the outcome variable was higher than the mean of the variable). In the first model (Table 4), all coverage types were entered as predictors, with newspapers as control variables. The model likelihood ratio is statistically significant, thus rejecting the null hypothesis that all regression coefficients in the model are simultaneously equal to zero, and the fit of the model is also confirmed by the Chi-Square/df ratio which is substantially higher than 0.05. We can also add that almost all residuals were under the absolute value of 2, whereas only a few cases were outliers with values slightly above absolute value of 2.0. From the results we can note that three out of four predictors were statistically significant. As for the balanced reporting, for every one-point increase in dissent, the rate of negative user comments increased by a factor of 1.69. As for the boomerang effect predictors, the presence of an explicit call for vaccination increases the number of negative comments by 1.76 times, explicit naming of the expert by 1.46 times, while the blaming of vaccine hesitant persons was not statistically significant when the other predictor variables are held constant.

In the second model (Table 5), we added interaction terms in order to test whether boomerang effects vary in different contexts, i.e., in relation to the presence of balanced reporting (scientific dissent). The results revealed that none of the three interaction effects was statistically significant, thus implying that there are no moderating effects. The model fit is also somewhat poorer in comparison to the first model.

## 4. Discussion

Overall, the results of our study revealed that online user comments are quite polarized and mostly skeptical about vaccination (RQ1). We also confirmed the balanced reporting effect (H1) and boomerang effect (H2) in COVID-19 reporting. The interactive effect (H3) was not confirmed.

Our findings show that negative sentiments are frequent in analyzed online user comments. These results are in line with the findings of previous studies in Croatia, which indicated that news related to the topics of vaccination, epidemic measures and COVID-19 daily reports are associated with fear [50]. Although it did not deal specifically with vaccination, an earlier Croatian study conducted in 2021 by Babić et al. [51] also showed that negative sentiment is dominant in tweets posted during the COVID-19 pandemic and findings of many other studies. A research study by Sufi et al. [52] found that anti-vax posts/tweets containing the word “hoax” had the highest level of social influence with the highest level of negativity. Another study found that anti-vaccination supporters are more engaged in discussions on Twitter [53]. We can conclude that our results are in line with similar studies [51,53,54] which have already found that negative attitudes and sentiments are dominant in social media posts during the COVID-19 pandemic, especially related to the topics of vaccination. Online user comments have the potential to strengthen pre-existing negative sentiments/attitudes [55] and public suspicions toward vaccines and be one of the important strongholds of the antivax movement.

Even though we applied the Hughes et al. coding scheme, our analysis showed that anti-vaccination tropes can be broadly divided into two types: vaccine-related and establishment-related [20]. In other words, even though Hughes et al. nuanced tropes and rhetorical strategies are valid, they are often employed together, and thus can be divided in the more general categories mentioned above [20]. Our study results show that in online user comments establishment-related tropes are much more frequent, thus confirming the issue of trust in the expert system that is vividly present throughout COVID-19 pandemics. Trust involves uncertainty and risk which are acknowledged and accepted as the part of the process of trust between competent and autonomous agents. The lack of trust in medical expert systems is not connected to the specific context of COVID-19 pandemic; it represents a long-term process of declining trust and criticism that emerged within the scientific community [56,57], mainly as the consequence of the risk societies development processes mentioned in the introductory part of this paper, wherein the trust in experts’ competence to control the risks is increasingly questioned [7,8]. Such a social environment questions “scientific monopoly” on the truth of the medical profession, which gradually loses its professional authority. However, it is important to note that majority of the scientists who were present in the Croatian media were members of various government/state bodies and that distrust towards the Government can be reflected in distrust towards experts/scientists. They are perceived as persons who are providing scientific legitimacy to government decisions without real scientific autonomy. In addition, study by Beliga et al. [50] have found that over time people become more dissatisfied with politicians and scientists as relevant actors in the COVID-19 pandemic. In this context, presence of negative sentiments in online user comments, as a reflection of distrust toward establishment/public officials, is not surprising.

Our results also revealed that balanced reporting has not been widely present during the COVID-19 pandemic in Croatia. Nevertheless, in cases where it was present, measured by the implied scientific dissent about COVID-19 vaccination, it elicited more negative reactions among the readers. Considering that, we can conclude that the influence of balanced reporting, i.e., our hypothesis H1 was confirmed. Therefore, the results of the current study cohere with the previously mentioned studies which determined that even small cues about scientific dissensus can make science attitudes more negative [31,32]. However, we have to bear in mind that this conclusion is based on the overall number of negative comments. Balanced reporting might have solidified the support for vaccination among less hesitant persons, which could not be measured within our research design.

The study results also partially confirmed the existence of the boomerang effect (hypothesis H2), given that two out of three boomerang effect predictors (presence of the explicit call for vaccination and naming of the expert) were significantly associated with more negative comments. The only predictor, which was not statistically significant, was blaming vaccine hesitant persons. This finding might be explained by the possibility that some of the readers did not pick up often subtle cues connected to the blame attribution. One of the possible explanations might lie in the fact that explicit calls for vaccination and named expert, as the other two measures predicting boomerang effect, are more often present in the article titles, thus being more accessible to the readers. Given the vast amount of information available on the Internet accompanying the current COVID-19 crisis, all the tropes that include scientists are known and readily available to the media users. Therefore, cues coming from assertive reporting are more likely to elicit negative reactions, especially in the context of the declining trust in expert systems explained earlier. It is particularly interesting that the very act of the presence of the concrete, named expert elicited more negative comments. It seems that concrete experts act as a more provoking signal for vaccine hesitant persons, probably enabling them to visualize “the enemy”. Here, we should also highlight that such a finding does not mean that experts do not have an overall positive role in communication about vaccines. This just means that it is possible that the use of experts in changing the minds of stubborn opponents of vaccination through direct dialogue might not be the best way. This is supported by a recent research study in Croatia, wherein the authors concluded that vaccination campaigns should not be focused only on the scientific arguments about safety and effectiveness of vaccine and on the authority of science and scientists because the study has shown that vaccine-hesitant groups do not trust scientists and experts [58]. However, this should be checked in future studies. Finally, we did not confirm our expectation that the boomerang effect will be stronger in cases where balanced reporting was present (hypothesis H3). A plausible explanation might be hidden in the fact that explicit calls for vaccination and placing the blame on vaccine hesitant persons very rarely went hand in hand with balanced reporting, i.e., implying scientific dissensus or mentioning vaccination harms. In other words, reporting that wholeheartedly recommended vaccination usually did not contain any challenges to the consensual view on the issue.

Overall, our study lends some support to the concept of reactance in health communication studies which, as Shoenberger, Kim and Sun noted, has been often referenced, but still not sufficiently tested [38]. The influence of the perception of scientific dissent on attitudes about vaccination can be reversed in two ways. The first is to undo the disagreement, but this is very difficult to achieve and it is questionable how desirable it is to do so. Grasswick suggests that the importance and impact of “scientific whistleblowers” stem from the failed expectations of the lay public, i.e., the expectations from the scientists to communicate and share significant knowledge [59]. Whistleblowers fill in the gap between science and the lay public, making the lay public feeling less marginalized. As Goldenberg notes, by failing to bridge the gap and to engage into meaningful conversation and to address the concerns which are relevant to the lay public, scientists fail to transfer the consensus that exists within the scientific community over the issues that worry the lay public and to earn epistemic merit and overall trust for pursuing public health measures [60]. As Aklin and Urpelainen suggested, probably the more promising venue is to try to “normalize” scientific disagreement, i.e., to prevent the relativization and politicization of science that occurs when even the smallest amount of uncertainty and scientific dissent is equated with the epistemological failure of science and the futility of following its public recommendations [32]. For example, Collins and Pinch [61] also argue that the public image of science as completely certain, neutral and objective is dangerous because it creates public expectations which cannot be fulfilled. If citizens are encouraged in such thinking, they will expect too much from science or be horrified by disagreement among scientists [62].

## 5. Conclusions

Our study extends previous research on balanced reporting and the boomerang effects by analyzing online new sites user comments as a potentially good approximation of the real-life discursive strategies of the pro-vaccination and anti-vaccination communication, thus being complementary to the mainly experimental research of the aforementioned effects. The results revealed that COVID-19 reporting was mainly consensual and pro-vaccination. However, user comments were highly polarized and mostly negative, with the majority of anti-vaccination tropes linked to the “corrupt elites”. We also established the negative influence of balanced reporting and the existence of a boomerang effect in cases of overtly persuasive affirmative reporting, thus confirming our main hypotheses and the experimental evidence from previous studies. The main general conclusion of our study is that the journalistic norm of balanced reporting does some damage to the credibility of science. However, aggressive communication based on the blaming and overtly persuasive communication can also elicit negative reactions, which hinder the achievement of the public health goals. The results of the study can be used for the adjustment of strategic communication, especially when communicating with vaccine hesitant audience. Based on this study’s results, it is recommended that the relativization and politicization of science should be prevented by not equating scientific consensus with an absolute epistemological certainty and by addressing legitimate concerns of vaccine hesitant persons without putting explicit blame on them.

As for the main limitation of the study, we should reiterate that the number of negative comments is not a good projection of the general attitudes, given that vaccination opponents can be assumed to be more motivated to respond to mostly positive vaccination coverage. Therefore, the conclusions of this study can hardly be generalized to the general population, but rather reflect the regularities that exist in the attempts to influence people with more negative attitudes about vaccination. Another limitation is relatively modest sample size (N = 300), which was dictated by the very strenuous and time-consuming human coding content analysis, which also included pre-testing and reliability analysis.

Based on the overall conclusions and limitations, future studies should aim to replicate the results using different indicators of balanced reporting and boomerang effects in order to increase the robustness of the results. In addition, it would be advisable to replicate the results on larger samples and in other countries.

## Figures and Tables

**Figure 1 vaccines-10-02085-f001:**
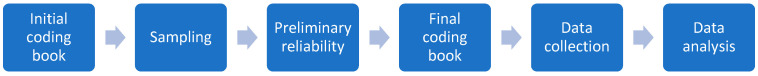
Research design.

**Figure 2 vaccines-10-02085-f002:**
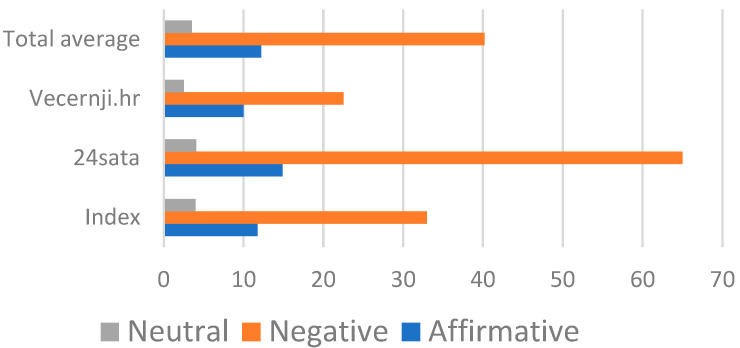
Comment types by online newspapers (average number).

**Table 1 vaccines-10-02085-t001:** Reliability measures.

Variable	Percentage Agreement	Krippendorff Alpha (Nominal or Interval)
Anti-vaccination tropes	87.34	0.63
Major antagonist	91.14	0.66
Anti-vaccination rhetorical strategy	83.54	0.70
Pro-vaccination tropes	91.14	0.79
Pro-vaccination rhetorical strategy	86.08	0.74
Number of negative comments	-	0.83
Number of neutral comments	-	0.88
Number of positive comments	-	0.79
Scientific dissent	-	0.81
Explicit call for vaccination	88.61	0.77
Blaming hesitant persons	92.41	0.82
Is the expert named	97.48	0.94

**Table 2 vaccines-10-02085-t002:** Descriptive statistics for tropes, antagonists and rhetorical strategies.

Variable	Answer Category	Percentage
Anti-vaccination tropes	Corrupt elites	69.67
	Vaccine injury	22.00
	None	8.33
Major antagonist	Mainstream society at large/politicians; shadowy villain	81.00
	Vaccine itself	11.00
	No clear antagonist	8.00
Anti-vaccination rhetorical strategy	The Brave Truthteller; Global Movements/Sleeping Giants	16.34
	Do Your Own Research; Mountains and Molehills	71.33
	No clear rhetorical strategy	12.33
Pro-vaccination tropes	Vaccines work/prevent diseases	64.67
	Medical workers as heroes	3.67
	Not available	31.67
Pro-vaccination rhetorical strategy	Rationalism against irrationalism	51.00
	Presenting evidence for vaccination	16.67
	Not available	32.33

**Table 3 vaccines-10-02085-t003:** Descriptive statistics for the characteristics of the issue coverage.

Variable	Answer Category	Percentage
Scientific dissent	Complete agreement	67.67
	Mainly agreement	15.67
	Both agreement and disagreement	9.67
	Mainly disagreement	3.00
	Complete disagreement	4.00
Explicit call for vaccination	Yes	54.00
	No	46.00
Blaming hesitant persons	Yes	74.33
	No	25.67
Is the expert named	Yes	71.24
	No	28.76

**Table 4 vaccines-10-02085-t004:** Negative binomial regression—model 1.

	B	S.E.	*p*	Exp (B)	95% Confidence Intervals for Exp (B)
Intercept	2.93	0.18	0.00	18.65	13.03–26.70
Newspapers = Index	0.26	0.15	0.09	1.30	0.96–1.75
Newspapers = 24 sata	1.00	0.15	0.00	2.72	2.02–3.66
Newspapers = Večernji list	-	-	-	-	-
Scientific dissent	0.52	0.07	0.00	1.69	1.48–1.92
Blaming = Yes	0.21	0.15	0.17	1.23	0.92–1.65
Explicit call = Yes	0.56	0.14	0.00	1.76	1.34–2.30
Main expert named = Yes	0.38	0.14	0.01	1.46	1.10–1.94

Outcome variable: count of negative user comments. Likelihood ratio χ2(6) = 158,03, *p* = 0.00; Pearson Chi-Square/df = 0.96; AIC = 2562.35; BIC = 2587.97.

**Table 5 vaccines-10-02085-t005:** Negative binomial regression—model 2.

	B	S.E.	*p*	Exp (B)	95% Confidence Intervals for Exp (B)
Intercept	1.24	0.30	0.00	3.49	1.93–6.30
Portal = Index	0.30	0.15	0.05	1.34	0.99–1.82
Portal = 24 sata	1.03	0.15	0.00	2.79	2.07–3.78
Portal = Večernji list	-	-	-	-	-
Scientific dissent	0.70	0.18	0.00	2.01	1.42–2.86
Blaming = Yes	0.63	0.26	0.02	1.88	1.13–3.12
Explicit call = Yes	0.56	0.28	0.04	1.75	1.02–3.02
Expert named = Yes	0.63	0.31	0.04	1.87	1.02–3.44
Blaming X Scientific dissent	−0.36	0.19	0.07	0.70	0.48–1.03
Explicit call X Scientific dissent	0.23	0.17	0.18	1.26	0.90–1.75
Expert named X Scientific dissent	0.03	0.23	0.89	1.03	0.66–1.61

Outcome variable: count of negative user comments. Likelihood ratio χ2(13) = 163.26, *p* = 0.00; Deviance/df = 0.97; AIC = 2563.12; BIC = 2599.72.

## Data Availability

Available on the journal’s website.

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
