# Peer review of "Balanced Reporting and Boomerang Effect: An Analysis of Croatian Online News Sites Vaccination Coverage and User Comments during the COVID-19 Pandemic"

_vaccines, 2022, doi:10.3390/vaccines10122085_

Round 1

Reviewer 1 Report

This paper was an interesting study. I have few comments:

1) Material & Method section is verbose. Please add one schematic diagram or flow chart of the overall methodology to clearly articulate the methodology.

2) There were few formatting / spelling issues. For example, in line 407, the first quotation mark preceding “corrupt” appears to be upside down. Similar issue is present in line 164, before the word balanced.

3) Indicate possible future work for this study within the conclusion section (after the limitations)

4) Some highly relevant recent studies on COVID-19 related vaccine sentiments should be referenced. The Author can add the following:

F. K. Sufi, I. Razzak and I. Khalil, "Tracking Anti-Vax Social Movement Using AI based Social Media Monitoring," in IEEE Transactions on Technology and Society, 2022, doi: 10.1109/TTS.2022.3192757. https://ieeexplore.ieee.org/document/9834043”

5) Moreover, within the discussion section, please compare your overall result with existing literatures on COVID-19 vaccine sentiment

6) Within the result section, it is advisable to add graphs/charts to overcome the monotonously verbose nature of the manuscript.

Author Response

Reviewer: This paper presents an interesting study. I have few comments:

Authors: Thank you for your effort and useful comments. The specific answers and changes which were made are indicated below.

Reviewer: Material & Method section is verbose. Please add one schematic diagram or flow chart of the overall methodology to clearly articulate the methodology.

Authors: We agree, at the begining of the methodology section now we added a flowchart which depicts research design, i.e. all the steps which had been taken. The details are described in the continuation, but the flowchart should make it easier to understand them.

Reviewer: There were few formatting / spelling issues. For example, in line 407, the first quotation mark preceding “corrupt” appears to be upside down. Similar issue is present in line 164, before the word balanced.

Authors: Thank you, we think this is corrected now.

Reviewer: Indicate possible future work for this study within the conclusion section (after the limitations)

Authors: Thank you for the suggestion, the following remarks are added to the closing section of the paper:

Based on the overall conclusions and limitations, future studies should aim to replicate the results using different indicators of balanced reporting and boomerang effects in order to increase the robustness of the results. In addition, it would be advisable to replicate the results on larger samples and in other countries.

Reviewer: Some highly relevant recent studies on COVID-19 related vaccine sentiments should be referenced. The Author can add the following:

“F. K. Sufi, I. Razzak and I. Khalil, "Tracking Anti-Vax Social Movement Using AI based Social Media Monitoring," in IEEE Transactions on Technology and Society, 2022, doi: 10.1109/TTS.2022.3192757. https://ieeexplore.ieee.org/document/9834043”

Moreover, within the discussion section, please compare your overall result with existing literatures on COVID-19 vaccine sentiment.

Authors: Thank you for the suggestion, in the discussion section we added a discussion of the connection between our results and vaccine sentiment research (they are in accordance), especially in Croatia. We also connected the results with the overall trust in healthcare profession and institutions.

Reviewer: Within the result section, it is advisable to add graphs/charts to overcome the monotonously verbose nature of the manuscript.

Authors: Thank you for this suggestion, we gave it a full consideration. We replaced Table 2 with figure, however Tables 3 and 4 comprised large amount of text and information, and we did not find an elegant way to present them in a figure which would be large and visible enough to convey the information.

Reviewer 2 Report

The manuscript was prepared very well. The introduction section justifies the purpose of the study. I congratulate the authors for the preparation of the manuscript

I would like to congratulate the authors for the structure of the manuscript and all the research carried out. It is highly publishable. However, there are some concerns, in part important, so the review articles need revision, see below.

Introduction

The introduction is too long, please making sections or summarize it. the reader has difficulty following the development of the introduction

The research questions and the study hypotheses could be included in the methodology and or left in the introduction, giving a redaction format.

Materials and Methods

The methodology is perfectly described and carried out

Results

·       The tables and the text describing them do not require any input, it is the strongest part of this study.

Discussion

·       include the results in the 1st paragraph in summary form and whether or not the hypotheses were met

·       what does this manuscript specifically contribute?

·       Include a limitations/strength section.

·       the authors should make their own assessment and include their own discussion of the results shown in the manuscript

·       In the Conclusion section, state the most important outcome of your work. Do not simply summarize the points already made in the body — instead, interpret your findings at a higher level of abstraction. Show whether, or to what extent, you have succeeded in addressing the need stated in the Introduction (or objectives).

Author Response

Reviewer: The manuscript was prepared very well. The introduction section justifies the purpose of the study. I congratulate the authors for the preparation of the manuscript.

I would like to congratulate the authors for the structure of the manuscript and all the research carried out. It is highly publishable. However, there are some concerns, in part important, so the review articles need revision, see below.

Authors: Thank you for your effort and kind remarks, below we provided the specific answers and the explanation of the changes that were made according to your comments.

Reviewer: The introduction is too long, please making sections or summarize it. the reader has difficulty following the development of the introduction. The research questions and the study hypotheses could be included in the methodology and or left in the introduction, giving a redaction format.

Authors: We agree. Therefore, we shortened the introduction by deleting information which was somewhat redundant and made the reading of the introductory part of the paper difficult.

Reviewer: The research questions and the study hypotheses could be included in the methodology and or left in the introduction, giving a redaction format.

Authors: We did this a full consideration, given the length of the introduction. However, the submission format does not accept a section entitled “research goals and hypotheses” which probably would be the best solution in this case. That is why we chose to keep the section within the introduction, where is probably better placed in comparison to “materials and methods”.

Reviewer: The methodology is perfectly described and carried out.

Authors: Thank you for this remark. According to the suggestion of the other reviewer, we added a flowchart in order to make the research design more vivid.

Reviewer: The tables and the text describing them do not require any input, it is the strongest part of this study.

Authors: Thank you for this remark. According to the suggestion of the other reviewer, we replaced one of the tables with a figure.

Reviewer: Include the results in the 1st paragraph in summary form and whether or not the hypotheses were met.

Authors: Thank you for this remark. Two sentences summarizing the study results are now added to the beginning of the section.

Reviewer: What does this manuscript specifically contribute?

Authors: Thank you for this suggestion , the following section describes the main contribution:

„Our study extends the previous research on balanced reporting and boomerang effects by analyzing online new sites user comments as a potentially good approximation of the real-life discursive strategies of the pro-vaccination and anti-vaccination communication, thus being complementary to the mainly experimental research of the aforementioned effects.“

Reviewer: Include a limitations/strength section.

Authors: This was already present to a degree, but now we slightly expanded it by adding suggestions for future research that could replicate this study in different ways.

Reviewer: The authors should make their own assessment and include their own discussion of the results shown in the manuscript

 Authors: The main assessment is given as follows:

„The results of the study can be used for the adjustment of strategic communication, espe-cially when communicating with vaccine hesitant audience. Based on the study results, it is recommended that the relativization and politicization of science should be prevented by not equating scientific consensus with an absolute epistemological certainty, as well by addressing legitimate concerns of vaccine hesitant persons without putting explicit blame on them.“

We also expanded the discussion of our results within the context of vaccine sentiment research and research about trust in scientists.

Reviewer: In the Conclusion section, state the most important outcome of your work. Do not simply summarize the points already made in the body — instead, interpret your findings at a higher level of abstraction. Show whether, or to what extent, you have succeeded in addressing the need stated in the Introduction (or objectives).

Authors: Thank you for this suggestion, we now added the generalization of the results in the following form:

„We also established the negative influence of balanced reporting and the existence of a boomerang effect in cases of overtly persuasive affirmative reporting, thus confirming our main hypotheses and the experimental evidence from previous studies. The main general conclusion of our study is that journalistic norm of balanced reporting does some damage to the credibility of science. However, aggressive communication based on the blaming and overtly persuasive communication can also elicit negative reactions which hinder the achievement of the public health goals.”

Round 2

Reviewer 2 Report

The authors have satisfied all my suggestions

I have nothing more to add